# Evaluation of routinely collected records for dementia outcomes in UK: a prospective cohort study

Shabina Hayat [1], Robert Luben,[2,3] Kay-Tee Khaw,[2] Nicholas Wareham,[2] Carol Brayne[1]

[1]Department of Psychiatry, University of Cambridge, Cambridge, UK
[2]MRC Epidemiology Unit, Cambridge, UK
[3]NIHR Biomedical Research Centre, Moorfields Eye Hospital NHS Foundation Trust & UCL Institute of Ophthalmology, London, UK

**Correspondence to**
Shabina Hayat;
sah63@medschl.cam.ac.uk

## ABSTRACT

**Objectives** To evaluate the characteristics of individuals recorded as having a dementia diagnosis in different routinely collected records and to examine the extent of overlap of dementia coding across data sources. Also, to present comparisons of secondary and primary care records providing value for researchers using routinely collected records for dementia outcome capture.

**Study design** A prospective cohort study.

**Setting and participants** A cohort of 25 639 men and women in Norfolk, aged 40–79 years at recruitment (1993–1997) followed until 2018 linked to routinely collected to identify dementia cases. Data sources include mortality from death certification and National Health Service (NHS) hospital or secondary care records. Primary care records for a subset of the cohort were also reviewed.

**Primary outcome measure** Diagnosis of dementia (any-cause).

**Results** Over 2000 participants (n=2635 individuals) were found to have a dementia diagnosis recorded in one or more of the data sources examined. Limited concordance was observed across the secondary care data sources. We also observed discrepancies with primary care records for the subset and report on potential linkage-related selection bias.

**Conclusions** Use of different types of record linkage from varying parts of the UK's health system reveals differences in recorded dementia diagnosis, indicating that dementia can be identified to varying extents in different parts of the NHS system. However, there is considerable variation, and limited overlap in those identified. We present potential selection biases that might occur depending on whether cause of death, or primary and secondary care data sources are used. With the expansion of using routinely collected health data, researchers must be aware of these potential biases and inaccuracies, reporting carefully on the likely extent of limitations and challenges of the data sources they use.

## INTRODUCTION

Routinely collected health records are powerful tools that have been increasingly used for research in the last decade.[1] Epidemiological studies depend on accurate, reliable and relevant information from data sources. However, there is substantial heterogeneity in the methods for case-ascertainment in dementia literature,[2] with studies applying different criteria, with no 'gold standard'.[3] Accuracy of medical records, of

---

## STRENGTHS AND LIMITATIONS OF THIS STUDY

⇒ This study has virtually complete follow-up of the cohort for over 25 years.
⇒ Exploring the complexity of dementia ascertainment in primary and secondary care settings.
⇒ An in-depth examination and new insight into the national mental healthcare data set.
⇒ Using medical records, a less sensitive approach for dementia diagnosis.

---

whatever type, tends to be assumed in the literature. Compared with population based surveys, dementia is still known to be under-reported in primary healthcare records, with individuals having manifest signs of dementia but no record of a dementia diagnosis, although this 'gap' is being closed more recently.[4] Dementia ascertainment is complex in primary and secondary care settings[5 6] as well on death certification.[7] There are many influences on whether dementia is likely to be recorded in these settings. Variations in dementia reporting have been noted across regions in UK.[8 9] Data sources are prone to inconsistency, misclassification and are influenced by change in dementia practice and policy,[8] such as those summarised in box 1.

In the UK, the National Health Service (NHS) holds data on all hospital admissions by condition, allowing dementia ascertainment from routine data sources. To date, mortality and Hospital Episode Statistics (HES) data have been the main sources of health records for epidemiological studies.[3 10 11] These data are coded using the International Classification of Diseases (ICD) which are mainly diagnostic codes. Using these sources alone seriously underestimates the number of dementia cases, as most dementia patients will not require hospitalisation for their dementia nor will they have dementia as a cause of death recorded on their death certificate.[10 12]

General practitioners (GP) maintain primary care records, which include information, both for diagnoses and also for administrative purposes, including details of specialist referrals

such as to memory clinics. Diagnosis of dementia is usually initiated in a primary care setting, based on patient symptoms or caregivers' concerns.[13] GP records, in theory, should be a more complete source for dementia case ascertainment. The coding system used in primary care uses Read Codes, is separate from the ICD system used in secondary care and death registration. Codes for dementia, are numerous, complex and prone to coding error and misclassification.[5] Another data source for dementia, which seems to have had less use in research, is the national mental healthcare data set (MHDS) which contain record-level data about individuals in contact with mental health services. The use of multiple systems across different settings and unfamiliar codes adds to the confusion of dementia case ascertainment.[5] The impact of incomplete or inaccurate recording of dementia can affect analysis and interpretation of results.[14] Furthermore, databases are influenced by the point data are collected in the patient care pathway[11] and may not be representative of the full population of people living with dementia, thus affecting generalisability of findings.[15]

Box 1 Changes in policy for public and practitioners which may have increased awareness of dementia and recording of cases in health records

**2006:** Introduction of dementia to Quality and Outcomes Framework (QOF) registers. QOF is an incentivising primary care scheme that rewards general practitioner practices for the provision of quality care and helps to standardise improvements in the delivery of primary medical services.[35] Participation in QOF is voluntary.

**2009:** The 2009 National Dementia Strategy was introduced to increase the public and professional awareness and understanding of dementia. Also, the dementia prevalence indicator to the QOF was introduced in primary care.

**2011:** Changes to coding practice. For mortality coding, deaths previously coded with an underlying cause of unspecified cerebrovascular disease were reclassified as vascular dementia.

**2012:** The Dementia Challenge was launched in March 2012 by Prime Minister, David Cameron. This programme's, which superseded the national strategy from 2009, focus was on three main areas: bringing about improvements in health and care, creating dementia friendly communities and improving research.

**2013:** The Dementia Commissioning for Quality and Innovation was introduced to encourage practitioners to identify those who potentially have dementia in a secondary care setting, most particularly applied to those over 75 years with unplanned hospital admissions.

**2015:** Prime Minister launched his Challenge on Dementia 2020, which set out to build on the achievements of the Prime Minister's Challenge on Dementia 2012–2015.

The objective of this study was to describe the procedures for a better understanding of the characteristics of individuals recorded as having a dementia diagnosis in different healthcare records and examine the extent of overlap of dementia coding across data sources. We present novel insight into the MHDS and compare secondary and primary care records providing value for others relying on these types of data sources for outcome capture.

## METHODS

### Study population and data collection

Participants were recruited to the European Prospective Investigation of Cancer (EPIC) in Norfolk. Details of this study have been previously described.[16] Briefly, 30 445 men and women (40–79 years) residents of Norfolk, England, were recruited to the study via their general practices. Of these, 25 639 attended the baseline health examination between 1993 and 1997. Subsequent follow-ups have involved self-report of health and lifestyle postal questionnaires and further health examinations.[17] Virtually complete follow-up for cohort for mortality and hospital admissions in EPIC-Norfolk has been established via linkage to routinely collected NHS databases in England (HES) and for mortality data for all participants using their unique NHS number and date of birth.

### Dementia ascertainment and diagnostic codes in hospital and death records and mental healthcare data

Data linkage to the EPIC-Norfolk cohort for NHS and mortality data was carried out by NHS Digital, a statutory body in England, permitted to receive identifiable patient data for linkage. The linked hospital records contain coded diagnostic information for all inpatient and day-case admissions.[18] We also obtained national MHDS, which contain record-level data about individuals in contact with mental health services including memory clinics.

Participants with incident dementia were defined as those free of dementia at the time of enrolment to the study and then identified with a dementia diagnosis recorded in medical records subsequently. For a subset, we compared the NHS Digital data sets of secondary care and mortality to GP records. GP records for individuals with a dementia diagnosis in secondary care but missing from data provided by practices were further reviewed by researchers in the third and final confirmation phase for any indication of a dementia code or mention in free text. Further details of the protocol and dementia codes are provided in the online supplemental information and table S1. QOF for dementia coding in primary care financially incentivises each recorded dementia diagnosis. Figure 1 is a diagrammatic representation of selection of participants and record linkage in the cohort.

### Ethics

EPIC-Norfolk has the permissions (Section 251 and where possible, explicit signed consent given by participants attending health examinations) to follow-up through medical record linkage. Procedures for this study have been approved by NHS Research Ethics Committee (NRES REC Ref: 98CN01). The Regulatory approvals from the Confidentiality Advisory Group (CAG) can be found at:

Section 251 Application number 059; https://www.hra.nhs.uk/planning-and-improving-research/application-summaries/confidentiality-advisory-group-registers/.

For the GP dementia case ascertainment and validation substudy—we also obtained approval from Norfolk &

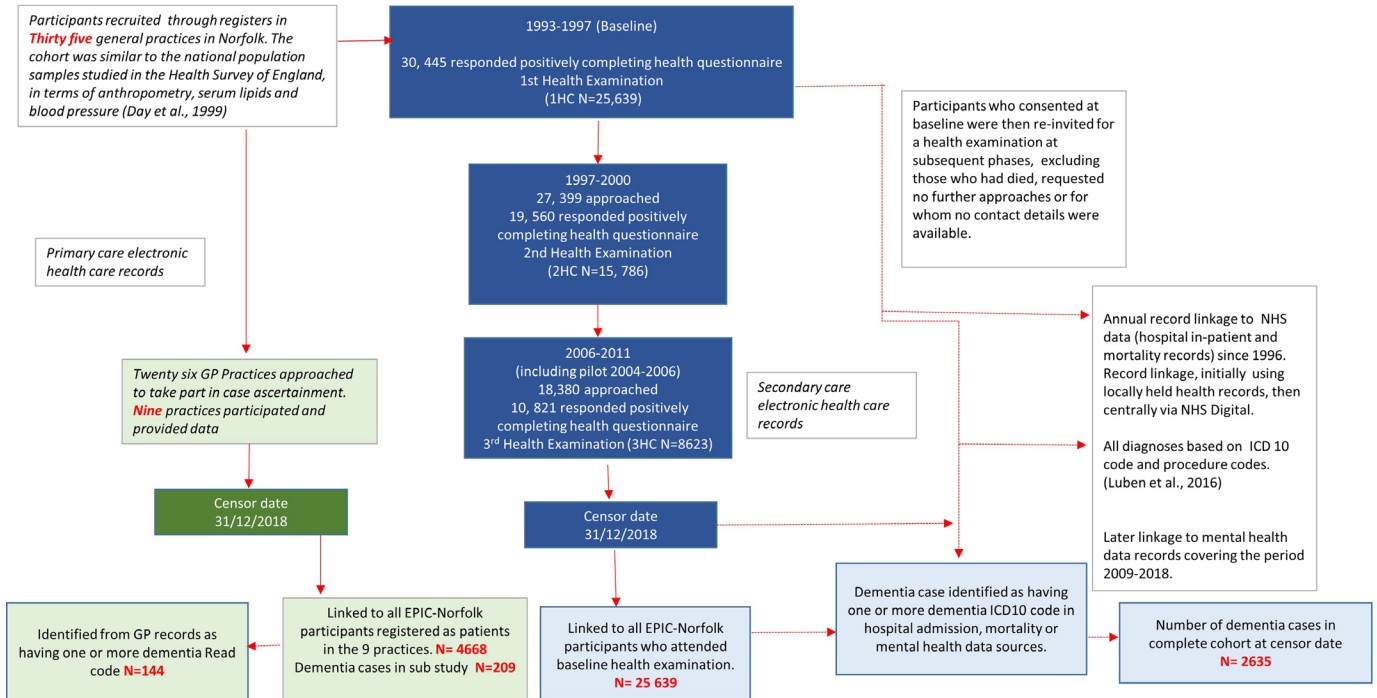

Key: Dotted lines represent passive follow-up and solid lines represent direct contact with GP Practice or participants

**Figure 1** Flow diagram of selection of EPIC-Norfolk participants for record linkage. EPIC, European Prospective Investigation of Cancer; ICD, International Classification of Diseases; NHS, National Health Service.

Suffolk Primary & Community Care Research Office for access to GP practices.

### Patient and public involvement

The EPIC-Norfolk study actively promotes greater participant involvement in research. The EPIC-Norfolk Participant Advisory Panel (EPAP) is a consultative forum providing participant perspective on how research is prioritised, planned, conducted, communicated and used. EPAP is viewed as a partnership between EPIC-Norfolk researchers and participants. Further details on EPAP can be found here: https://www.epic-norfolk.org.uk/for-participants/epap/

### Statistical analyses

Definite dementia cases were defined as having one or more of the ICD-10, or (for GP records) Read code, for dementia. Age and sex-specific incidence rates for all-cause dementia and mortality rates were calculated for the entire cohort. The number and proportion of participants with a diagnosis of dementia from the three NHS Digital data sources separately and combined were examined. We also examined differences in sociodemographic characteristics (age at recruitment, sex, education and social class) of dementia cases across the three NHS Digital data sets. Details on covariates are provided in the online supplemental information. We used Cox proportional-hazard models to compare the association of sociodemographic factors and risk of a dementia diagnosis across data sources, with mutual adjustment of covariates in the model. Statistical analyses were performed using SPSS V.25.0 (IBM Corp., Armonk, NY, USA).

### RESULTS

There were 2635 cases of dementia identified from the cohort of 25 639 individuals at the censor date of 31 December 2018 after 25.8 years of follow-up. The youngest age of entry to the study at baseline was just below 40 and the oldest age of the participant at the censor date was 101 years. Out of the 2298 individuals with data on age of diagnosis, the minimum age of diagnosis was 54 years and maximum was 99 years. Figure 2 shows the relationship between age and dementia diagnosis from mid to later life. Table 1 shows the sex-specific and age-specific cumulative incidence of dementia and deaths in the cohort. Increasing age was associated with increasing rates of dementia and death. This table reflects

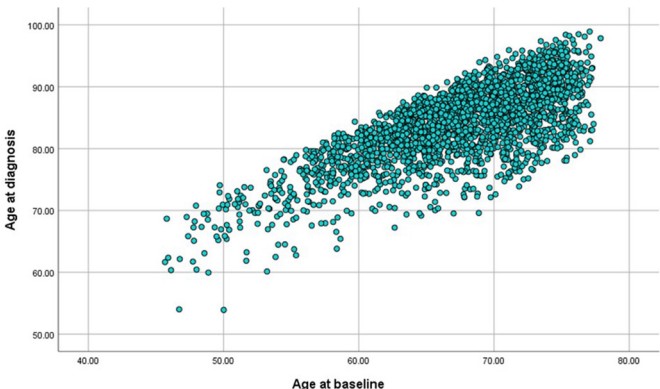

**Figure 2** Scatter plot of age of diagnosis by age at baseline in the EPIC-Norfolk cohort. EPIC, European Prospective Investigation of Cancer.

**Table 1** Age-specific and sex-specific proportions of dementia and death in EPIC-Norfolk from 1996 until 31 December 2018 using all three secondary care data sources provided by NHS digital for each age group by gender at baseline

| Age band at baseline | Median age | Freq (N) | % with dementia recorded at any time in follow-up (N) | % Died (N) | P value |
|---|---|---|---|---|---|
| Men (n=11 607) | | | | | |
| ≤59 years | 51.6 | 5915 | 3.2 (190) | 17.4 (1028) | <0.001 |
| 60–64 years | 62.5 | 1848 | 12.3 (228) | 46.7 (863) | |
| 65–69 years | 67.5 | 1890 | 15.6 (294) | 72.0 (1361) | |
| 70–74 years | 72.5 | 1579 | 17.3 (273) | 89.9 (1420) | |
| >75 years | 75.7 | 375 | 14.9 (56) | 95.7 (359) | |
| Women (n=14 032) | | | | | |
| ≤59 years | 51.3 | 7656 | 2.9 (222) | 11.9 (909) | <0.001 |
| 60–64 years | 62.6 | 2118 | 14.5 (308) | 36.9 (782) | |
| 65–69 years | 67.4 | 2103 | 22.6 (475) | 55.4 (1166) | |
| 70–74 years | 72.4 | 1766 | 27.0 (477) | 80.4 (1419) | |
| >75 years | 75.8 | 389 | 28.8 (112) | 90.7 (353) | |
| All (n=25 639) | | | | | |
| ≤59 years | 51.5 | 13 571 | 3.0 (412) | 14.3 (1937) | <0.001 |
| 60–64 years | 62.6 | 3966 | 13.5 (536) | 41.5 (1645) | |
| 65–69 years | 67.5 | 3993 | 19.3 (769) | 63.3 (2527) | |
| 70–74 years | 72.5 | 3345 | 22.8 (750) | 84.7 (2839) | |
| >75 years | 75.7 | 764 | 22.0 (168) | 93.2 712) | |

P values by $\chi^2$ for proportion of individuals with dementia in each age category. Results shown for men and women separately and combined (all).
EPIC, European Prospective Investigation of Cancer; NHS, National Health Service.

the higher mortality in men which results in higher absolute numbers and of dementia cases in women.

On comparing characteristics across the different data sources, there were no major differences in terms of sex and sociodemographic profiles (table 2). Those with dementia recorded from the mortality data set were slightly older and in MHDS were younger. Most recorded diagnoses of dementia were found in HES, followed by mortality records, with the least number identified from the MHDS. The distribution of dementia diagnoses across the three NHS Digital data sources is shown in figure 3. This figure shows very little overlap. Where there was concordance in cases for the MHDS and HES data sources (n=448), 71% (319) had a date of diagnosis in the MHDS before the date of diagnosis in HES (median (IQR) 0.7 years (0.3, 1.7 years) with 26% (116) with an earlier date in HES and median (IQR) 0.6 years (0.2, 1.9 years). Only 3% (13) had the same date recorded in both.

Associations for sociodemographic factors and the presence of a recorded diagnosis of dementia were similar across the three data sources (table 3). Age was a stronger predictor in the mortality data and weakest in the mental health data. Having qualifications was associated with a lower risk of future recorded diagnosis of dementia, this was observed for mortality and HES data but not for the MHDS.

The MHDS consisted of mainly administrative data such as mental health reviews, care programmes and pathways that include contacts with mental healthcare professionals (both in hospitals and in outpatient memory clinics and the

community) as well as diagnostics and treatment codes. The service-level breakdown of the mental health data was not applicable here, as there was little additional diagnostic information. The latest release in 2017–2018 appeared to be the most complex, covering mental healthcare more comprehensively and containing diagnostic ICD10 codes that had been limited in the previous years.

For the substudy involving validation with GP records, out of the 26 practices that were contacted, 14 agreed to participate, 6 declined and 6 did not respond. Of the 14 that agreed, 8 practices completed the questionnaire and 9 provided data. There were no criteria for selecting a practice for this validation, other than they were practices that were collaborating with the study. There was a mix of rural and city practices, classified as urban or town and fringe areas,[19] although the majority of the practices ultimately submitting data were city-based. Different types of staff carried out the linkage with their records, ranging from practice managers, IT managers, research nurses to a healthcare assistant.

In this subpopulation, 4.4% (209 of the 4668 participants) were dementia cases identified from HES, mental health data, mortality and the GP records compared with 10.1% (2635) dementia cases out of 25 635 participants in the rest of the cohort. The individuals of the substudy registered with the responding practices were younger, more likely to be women, with higher education and social class when compared with the rest of the cohort (online supplemental table S2). Of the 209 respondents with a recorded diagnosis of dementia, 57

**Table 2** Comparison of characteristics of 'definite dementia' cases identified from the three data sources separately and combined

| | All three data sources HES/MHDS/mortality | | HES | | MHDS | | Mortality | | |
|---|---|---|---|---|---|---|---|---|---|
| | Definite dementia | | Definite dementia | | Definite dementia | | Definite dementia | | P value |
| | n=2635 | | n=2157 | | n=727 | | n=1276 | | |
| Sociodemographic | | | | | | | | | |
| Mean (SD) | | | | | | | | | |
| Age at baseline | 66.7 | (6.5) | 66.9 | (6.3) | 64.1 | (6.5) | 67.9 | (5.9) | <0.001 |
| Age at diagnosis | 83.8 | (6.5) | 83.7 | (6.5) | 82.7 | (6.8) | 84.6 | (6.4) | <0.001 |
| Sex, % women (n) | 60.5 | (1594) | 59.8 | (1290) | 59.7 | (434) | 60.5 | (772) | 0.9 |
| Education, % (n) | | | | | | | | | |
| No qualifications | 48.4 | (1274) | 49.1 | (1057) | 45.3 | (329) | 50.4 | (641) | 0.5 |
| O/A level standard | 42.6 | (1120) | 41.9 | (903) | 45.3 | (329) | 41.4 | (527) | |
| Graduate level | 9.0 | (238) | 9.0 | (194) | 9.4 | (68) | 8.2 | (105) | |
| Social class, % (n) | | | | | | | | | |
| Professional | 6.3 | (162) | 6.3 | (132) | 5.6 | (42) | 6.0 | (74) | 0.7 |
| Managerial | 33.9 | (866) | 33.6 | (702) | 32.0 | (240) | 34.0 | (421) | |
| Skilled non-manual | 20.5 | (523) | 20.5 | (428) | 21.1 | (158) | 21.7 | (268) | |
| Skilled manual | 21.3 | (544) | 21.2 | (442) | 21.4 | (160) | 20.5 | (253) | |
| Semi-skilled | 14.2 | (362) | 14.1 | (295) | 17.2 | (129) | 13.6 | (168) | |
| Non-skilled | 3.8 | (98) | 4.2 | (88) | 2.7 | (20) | 4.3 | (53) | |

HES, Hospital Episode Statistics; MHDS, mental health data set; N, number; O/A, ordinary or advanced level.

were found in both secondary and primary care data and 87 were in primary care records only. In this small study, almost all the practices were using Read codes and all reported their dementia cases to be confirmed through secondary referral to memory clinics. Five out of the eight practices reported that some dementia cases were not previously identified and so were missing on their QOF registers. The summarised responses to the questionnaire from GPs are given in online supplemental information table S3. Participants who were prescribed one of the four dementia drugs (n=57), also had a Read code of a definite dementia diagnosis. There were no cases identified from drugs alone.

There were 65 participants who had a definite dementia diagnosis in the secondary care records, but not in their GP records. For most of these (51 cases), the reason for absence from GP systems was because the participant had died, and the dementia diagnoses had come from mortality records. For the remaining (n=14), 3 participants had Mild Cognitive Impairment or probable dementia and for 10 participants, we were unable to confirm diagnosis. This could be because they had died recently, or had moved to another practice, in which case their records would no longer be available to the GP. One patient had no indication of any dementia or dementia related condition in their record. Of the 57 cases that were in both the secondary and primary data sources, 30 cases were obtained from the initial GP data extract, with 27 missed by the practices. These cases were ascertained with further review of individual records by the researchers and would have been missed if the extra confirmation phase had not been a part of the study design. Distribution of dementia cases across primary and secondary care data sets are shown in figure 4.

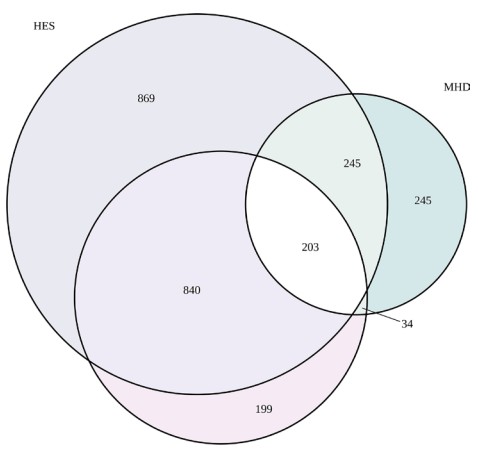

**Figure 3** Distribution of cases (n=2635) across the three main data sources from NHS digital followed up until 31 December 2018. HES, Hospital Episode Statistics; MHDS, mental health data set; NHS, National Health Service.

## DISCUSSION

Medical records allow virtually complete follow-up for dementia research, however there are limitations in terms of accuracy, completeness and underestimation of cases across

**Table 3** Association of sociodemographic factors and dementia in the EPIC-Norfolk cohort in the three data sources combined and individually, mutually adjusted for age, sex, education and social class

| Factor | All three data sources (dementia=2555) | | | HES only (dementia=2087) | | | MHDS only (dementia=715) | | | Mortality only (dementia=1229) | | |
|---|---|---|---|---|---|---|---|---|---|---|---|---|
| | HR | 95% CI | P value | HR | 95% CI | P value | HR | 95% CI | P value | HR | 95% CI | P value |
| Age per 5 years | 2.19 | (2.13 to 2.26) | <0.001 | 2.23 | (2.16 to 2.31) | <0.001 | 1.80 | (1.71 to 1.89) | <0.001 | 2.57 | (2.46 to 2.69) | <0.001 |
| Sex (men) | 0.98 | (0.91 to 1.06) | 0.6 | 1.01 | (0.92 to 1.10) | 0.9 | 1.03 | (0.88 to 1.20) | 0.7 | 1.08 | (0.96 to 1.21) | 0.2 |
| Education (any qualifications) | 0.87 | (0.80 to 0.95) | 0.002 | 0.87 | (0.79 to 0.95) | 0.002 | 0.91 | (0.78 to 1.07) | 0.2 | 0.84 | (0.74 to 0.94) | 0.003 |
| Social class (non-manual) | 0.99 | (0.91 to 1.08) | 0.8 | 0.98 | (0.89 to 1.07) | 0.7 | 0.93 | (0.79 to 1.09) | 0.4 | 1.03 | (0.91 to 1.16) | 0.7 |

EPIC, European Prospective Investigation of Cancer; HES, Hospital Episode Statistics; MHDS, mental health data set.

data sources.[3 6 11 20 21] This study confirms that, although there is some concordance for dementia across data sources, this is rather limited. Hospital records yielded the greatest number of cases, followed by death certification with the least number identified from the MHDS.

As in other studies,[3 10 22] we interpreted the absence of a dementia diagnosis code as absence of the dementia; this is clearly not the case. Even though using medical records is less sensitive to an in-study algorithmic approach to dementia diagnosis, specificity in all these data sources is likely to be high, as a clinical diagnosis, particularly in primary care records is usually made after referral to a specialist.[23] Another weakness is that we did not inspect medical records for the entire cohort, and so likely to have missed cases, reducing sensitivity. Furthermore, we would have missed milder cases as we only included definite dementia diagnosis, other than when confirming cases, where we did search for milder impairment.

To maximise our outcome and minimising risk of misclassification, we did not analyse types of dementia separately. Misclassification is common, and the emphasis on clinically diagnosed Alzheimer's disease (AD) in the majority of individuals hides the fact that most people with dementia, namely those aged 80 and above have mixed pathologies in their brains. Those who die without dementia also have increasing expression of AD pathologies.[24 25] Treating dementia as a single entity at a population level is appropriate as many risk factors are shared. Misclassification may have some impact on associations for younger onset dementias, where the level of importance of this will depend on the purpose of the study. Using a broader classification also improves external validity.[26] Underascertainment is inevitable and should be considered when making prevalence estimations.

Another limitation is that we did not account for the competing risk of death, which will be high in this ageing population. However, the proportion of deaths by age in men and women were presented. Individuals reaching the end of life with other comorbidities may die before any diagnosis of dementia.[27] To account for the competing risk for death, death must be a discrete event from dementia, and given that we included dementia from mortality records, this overlap does not allow for competing risk of death to be estimated here. However, in the Cox regression models, these individuals would have been censored, and once censored, these individuals are no longer at risk.

Using simple sociodemographic factors to examine associations with future risk of a definite dementia diagnosis, we demonstrate small differences, across the main two data sources, HES and mortality, with more difference observed in the MHDS. The age differences observed across the three data sources is as anticipated, as younger individuals with memory concerns are more likely to be referred to memory clinics compared with older frailer individuals who are more likely to be identified through hospitals.

The lower risk of dementia diagnosis for those with qualifications observed for mortality and HES data but not for the MHDS, could be due to the smaller numbers

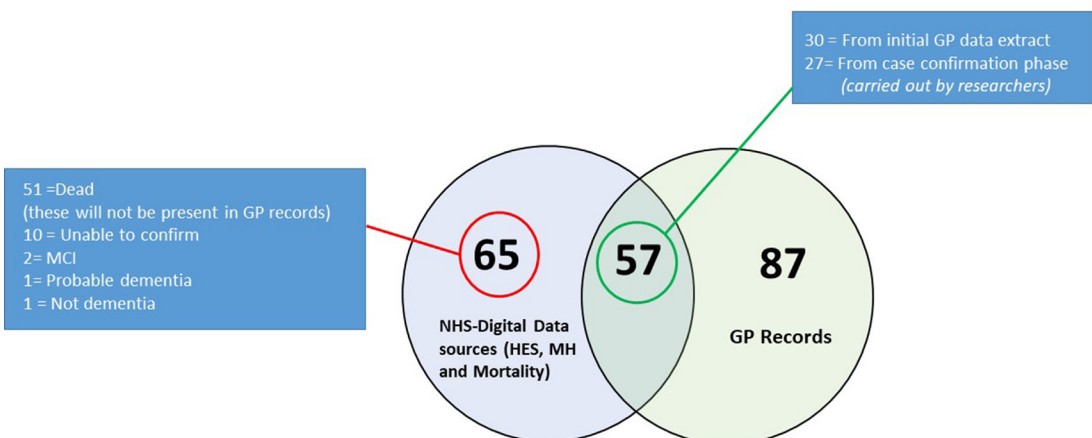

**Figure 4** Distribution of cases (n=209) identified from primary and secondary care in subpopulation of EPIC-Norfolk participants (n=4668). EPIC, European Prospective Investigation of Cancer; GP, general practitioners; HES, Hospital Episode Statistics; MCI, Mild Cognitive Impairment; MH, mental healthcare; NHS, National Health Service.

in the MHDS, or reflecting that there was no difference in terms of education as to who accesses mental health services. For the sociodemographic factors examined here, we had few missing data, however for more complex factors where there is likely to be more missing data, the potential for these differences to be greater. Such factors and unrecorded dementia cases in a data source could yield different study results. Studies have highlighted regional variation in rates of diagnosis and reliability of existing data.[3] We have demonstrated that this heterogeneity exists even in a single geographical region.

We were also able to draw on the relatively new MHDS from NHS Digital, as yet not widely used in research or described in detail. Where there was concordance in cases for the MHDS and HES data, we found that for most cases, the date of diagnosis was earlier in MHDS by just a few months. This is as expected, as the MHDS will be from referrals made from GP, where diagnosis of dementia is usually initiated by family or individuals themselves.[13] All activity relating to patients receiving care for a suspected or diagnosed mental health, learning disability or neurodevelopmental conditions is within scope of this data set. This is a further source of clinical and operational data in the NHS that can be used for purposes other than direct patient care. Although the MHDS covers mental healthcare more comprehensively than the other sources, it is limited as most of the clinical information is not coded, but recorded in text, thus, not lending itself so easily large-scale analyses.[7]

We found MHDS to be complex, with information on individuals (from referral to final discharge) on contact with secondary mental health services. Each subsequent annual release of the mental health data was wider in scope than the previous version. The data in the final year of follow-up gave the greatest number of dementia cases by ICD code. The sharp increase in diagnosis codes shows how important understanding the policy and practice context is to assess the influences and potential biases inherent in use of routine data sources. Additional reasons include changes in coding practices in mental

health services, or the way these data are extracted to include diagnoses as well as service codes. The MHDS has far greater potential to provide more complete estimates of diagnosed dementia in the population in the UK. This data set will need further reviewing for future work.

It is currently not possible to examine primary care records via databases such as the Clinical Practice Research Datalink (CPRD) for this cohort. As we had to rely on GP practices to extract the data, we were restricted, for practical reasons, to link to a subset only. The protocol was time and labour intensive. We did not observe the same level of concordance with hospital records as shown in other larger cohort studies.[10 22] Some of this discrepancy was due to patients having died as GP records for people who die are moved swiftly to NHS archives. Due to financial incentivisation, there has been an increase in dementia diagnosis in primary care, although concerns of overdiagnosis have been raised and should be monitored carefully.[4] Diagnosis of dementia is usually initiated in a primary care setting and could be considered to be the most complete single source of dementia case ascertainment, but underdiagnosis still exists[15 28 29] for various reasons including reluctance of GPs to diagnose dementia.[29]

As with the UK Biobank study,[22] we also found significant proportion of dementia cases in primary care records, that had not been found in hospital or mortality records. In EPIC-Norfolk, this figure was 42% (87 out of the 209 dementia cases identified) compared with 52% reported in UK Biobank. The lower proportion in our study reflects the older age of our cohort, and so more likely to appear in hospital and mortality records than the participants from UK Biobank. UK Biobank included participants from Scotland where data quality is high and centralisation of data, linkable health service data sets make data more accessible to research. This position is not currently shared by the other countries of UK, although may change in the near future with the establishment of Health Data Research UK (HDR-UK) the vision to improve population health, address health inequalities and to drive efficient service provision.[30]

Another large cohort study, the Million Women Study, found NHS hospital admission data to agree with primary care records.[10] This study only consisted of women, and also like UK Biobank, participants were younger.

The study in the subpopulation of the EPIC-Norfolk cohort with GP records has highlighted several key issues. Even though previous work has shown that practices are confident in identifying dementia cases,[9] the complexity of 'missingness' is well recognised.[31] If we had relied on the initial data extract from practices, we would have missed almost half the dementia cases. These cases were only ascertained via the confirmation step which involved a more detailed interrogation of the medical records by the research team. This may be due to lack of technical capacity or capability and could be a further reason for underascertainment. This reveals the potential scale of missed outcomes where known cases are not reflected in extraction of routine records, influencing studies using databases such as CPRD.[32] These findings highlight the importance of extending further training and knowledge of dementia coding in primary care.

It is clear from the characteristics and lower dementia incidence that the subpopulation was different from the overall cohort with regard to age, education and social class and therefore assuming generalisability to wider populations requires appropriate caveats. Although, there were no specific criteria set in approaching practices, the practices that responded were mainly city practices, and the patient-base in these practices was younger, more educated and of higher social class. It is likely that the more research-active practices responded. This bias is clearly reflected in the dementia rates, which was 10% in the EPIC-Norfolk cohort overall, compared with 4% in the subset from the nine GP practices that finally took part.

Whether secondary care data add any further to what can be found in GP record could not be explored in our small study. There were 10 additional participants with a record of dementia in cases from secondary care records, although we were unable to confirm these diagnoses as we could not access their records to check on the detailed information. One additional participant had no indication of dementia or dementia-related condition in the GP records despite a HES record of dementia. This may be a more widespread issue and should be explored in future work involving a larger and wider range of practices.

Databases such as CPRD are extremely powerful data sources, but rely on how dementia is coded in primary care.[5] Furthermore, they are not representative of all practices in the UK based on geography and size.[31] Our findings reveal the potential scale of biases influencing studies using larger primary care databases. They also highlight the importance of extending further training and knowledge of dementia coding in primary care. HES and mortality also rely on data provided by practices. Our findings show the extent to which practice selection impact on dementia case ascertainment and thus on measured rates. Therefore, these data should be used and interpreted with caution.

Recently, there have been changes to coding within primary care with the introduction of new codes SNOMED CT.[33] This new coding system will replace Read codes and eventually also be used in secondary care providing clarity and consistency. However, the implementation of this will take several years and the impact of this is unknown. There is also potential in the mental health data that includes other service-related codes that could be utilised for further insight into the level of cognitive impairment in the community. There are several administrative codes that relate to low, moderate and severe cognitive impairment. This information could be used to supplement the diagnostic information and could be useful in ascertaining milder forms of impairment and dementia.

This study provides confidence that identification of cases via record linkage with hospital admission, mortality and primary care data is sufficient for epidemiologic analyses of risk factors of dementia. The reliability of these data for incidence and prevalence rates is more challenging due to variability in ascertainment and diagnostic criteria which may differ over time and in different populations. In summary, it is important to note that different data sources provide different information. Using a single data source would clearly underestimate dementia outcomes, and so drawing from multiple sources is the best approach to maximise dementia ascertainment from routinely collected health records[3 15] and to enhance generalisability.[15] Researchers must be fully aware of the strengths and limitations of the data sources they use, identifying the potential sources of bias[34] and be transparent in reporting on how these reflect on the accuracy of their findings.

**Acknowledgements** The authors would like to thank all study participants, general practitioners and the EPIC Norfolk study team for their contribution to this work. We would also like to thank Nichola Dalzell who played a huge part in the data collection phase of the study.

**Contributors** SH drafted the manuscript, undertook analysis of the data and is the guarantor for this manuscript. SH was the lead applicant for acquiring the NHS Digital data for health and mortality endpoints, including the mental health data set, designed and conducted the case ascertainment, carried out the mapping of dementia codes and data cleaning. RL created the data set, carried out the data linkage and provided the graphics for Figure 3. K-TK, NW and CB are principal investigators who contributed to the conception and study design and guidance on the presentation and interpretation of these data. All authors contributed to data analysis and interpretation of the data.

**Funding** The EPIC-Norfolk study (DOI 10.22025/2019.10.105.00004) has received funding from the Medical Research Council (MR/N003284/1 MC-UU_12015/1 and MC_UU_00006/1), Cancer Research UK (C864/A14136) and NIHR https://www.nihr.ac.uk (Ref: NF-SI-0616-10090 to [CB]).

**Competing interests** None declared.

**Patient and public involvement** Patients and/or the public were involved in the design, or conduct, or reporting or dissemination plans of this research. Refer to the Methods section for further details.

**Patient consent for publication** Consent obtained directly from patient(s).

**Ethics approval** This study involves human participants and was approved by NHS Research Ethics Committee (NRES REC Ref: 98CN01). Participants gave informed consent to participate in the study before taking part.

**Provenance and peer review** Not commissioned; externally peer reviewed.

**Data availability statement** Data are available upon reasonable request. Data are available under a Data Transfer Agreement to any bona fide researcher. Although the data set is anonymised, the breadth of the data included and the multiplicity of

variables in the analysis file as primary variables or confounding factors means that provision of the data set to other researchers without a Data Transfer Agreement would constitute a risk. The contact for data request is: E-mail: EPIC-Norfolk@mrcepid.cam.ac.uk.

**ORCID iD**
Shabina Hayat http://orcid.org/0000-0001-9068-8723

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
