## [Reviewer comments · BMJ Open]

ARTICLE DETAILS

TITLE (PROVISIONAL)	Evaluation of routinely collected records for dementia outcomes in United Kingdom: a prospective cohort study.
AUTHORS	Hayat, Shabina; Luben, Robert; Khaw, Kay-Tee; Wareham, Nicholas; Brayne, Carol

VERSION 1 – REVIEW

REVIEWER	Charles Marshall Queen Mary University of London, Preventive Neurology Unit
REVIEW RETURNED	01-Feb-2022

GENERAL COMMENTS	This is a really useful paper for those working in the epidemiology of dementia, because it allows the comparison of dementia case ascertainment from major sources of records data in the UK. The major limitation is the small number of GP practices responding to the primary care component of the study. This is appropriately addressed by the authors, but makes it difficult to reliably compare the dementia ascertainment in primary care. I have only a few minor comments: 1. QOF stands for Quality and Outcomes Framework (not Quality of Framework as at start of Box 1)2. It might be worth clarifying that the QOF for dementia coding in primary care directly incentivises each recorded dementia diagnosis, leading to a large jump in ascertainment through primary care records from this time (and probably therefore making them the best single source of diagnosis in this type of study)3. Related to the above, even in the small sample of GP practices here, primary care records do seem to be the best data source especially for those who are still alive, and it might be worth bringing this out a bit more in the discussion. As someone who has worked with death records and primary care data, my main take away message from the paper is that primary care records are likely to be the best source for future work.4. Figure 3 is the main crux of the paper. Is there any way it could be drawn so that the size of the areas in the Venn correspond to the number of cases? This would make it easier to interpret visually.
--

REVIEWER	Kelvin Jordan Keele University, School of Primary, Community and Social Care
REVIEW RETURNED	15-Feb-2022

GENERAL COMMENTS	Studies in dementia are increasingly using electronic health records, however concerns are raised about whether cases are missed giving
---

	the complexity of diagnosis and recording. This study compared primary care, secondary care, mortality and mental health registers on diagnosed cases of dementia. The strengths are the linkages across these datasets including MHDS. The weakness is the low number of practices included. 1. The key question it seems to me given dementia is managed in primary care and researchers will generally start with primary care records is how good GP records alone are in identifying dementia. Mortality and hospital records are limited as not all patients will have died and not all patients will attend hospital due to dementia (did you use secondary as well as primary diagnoses?). So the question for the other sources is then really about whether they add patients missed in GP records. Figure 4 seems somewhat reassuring in terms of completeness of GP diagnoses of dementia given the biggest group missed are those who have died who did not still have their records in primary care. This assessment though is weakened by low number of practices taking part so there may be an issue of generalisability, particularly if the best coding practices took part. 2. I was slightly confused by P.9, line 10 "Participants with a dementia diagnosis in secondary care, but not indicated in the GP data were further reviewed by researchers in the third and final confirmation phase". What was the evidence they were looking for, if not a Read / SNOMED code, and has this been validated as a means of identifying people with dementia? Was it a free text mention of dementia? This information is important for researchers wanting to use primary care records to know how to obtain all cases of dementia (particularly as free text usually isn't available). 3. It would be useful to know, where there is concordance, what impact does using one source rather than another have on date of diagnosis – which is important for both research using dementia as an exposure or as an outcome. Mortality data obviously would not be used for the former but it is possible dementia was undiagnosed until death? 4. Is concordance of dementia diagnosis across data sources improving over time? Recruitment goes back to 1993 when recording quality was presumably lower. 5. I understand why the researchers did not explore sub-types of dementia but it is worth discussing a) whether issues affect specific types more, b) potential for people to be recorded with different types over time, or the incorrect type. 6. I wonder if some codes have been missed eg E00.. Senile/presenile dementia; F116. Lewy body disease; F111. Pick's disease 7. Table 1 – why don't the numbers for men and women with dementia diagnosis within an age band sum to the total for that age band? 8. I don't think figure 2 adds anything and could be deleted. 9. I was not sure of the purpose of the analysis behind table 3. Socio-demographic characteristics associated with dementia seems a different question to the underling question here around differences between the sources.
--	---

VERSION 1 – AUTHOR RESPONSE

Reviewer: 1 Dr. Charles Marshall, Queen Mary University of London

Reviewer's comment: The major limitation is the small number of GP practices responding to the primary care component of the study. This is appropriately addressed by the authors, but makes it difficult to reliably compare the dementia ascertainment in primary care.

Authors' response: We thank Dr Marshall for his comment. We agree that the small numbers included in the primary care component does make it difficult to compare with other studies, however, our findings are still widely applicable. Databases such as the Clinical Practice Research Datalink (CPRD) are more powerful resources but, as we mentioned in the manuscript, we were not able to link to our cohort at this time. Using the direct approach with the live records in GP Practices as we did, resulted a smaller but very comprehensive primary care component. The main purpose of this component was to provide a more detailed qualitative insight into dementia recording in primary care. The key message here is that even in this smaller study, we are able to give insight in terms of missing data in primary care records that will have a direct impact on the reliability on large primary care databases. We felt this is an important aspect of the study, given the increasing ambition to use routine primary care data by researchers.

Reviewer's comments (minor)

1. QOF stands for Quality and Outcomes Framework (not Quality of Framework as at start of Box 1)

Authors' response: We thank Dr Marshall for highlighting this mistake and have corrected in Box 1 and in the main text under 'Dementia case ascertainment through GP records' section

2. It might be worth clarifying that the QOF for dementia coding in primary care directly incentivises each recorded dementia diagnosis, leading to a large jump in ascertainment through primary care records from this time (and probably therefore making them the best single source of diagnosis in this type of study)

Authors response: We have now added that QOF is an incentivising primary care scheme (in Box1, page 7 of tracked version of manuscript) and stated in the methods that QOF for dementia coding in primary care financially incentivises each recorded dementia diagnosis.

3. Related to the above, even in the small sample of GP practices here, primary care records do seem to be the best data source especially for those who are still alive, and it might be worth bringing this out a bit more in the discussion. As someone who has worked with death records and primary care data, my main take away message from the paper is that primary care records are likely to be the best source for future work.

Authors' response: We have shown that the GP records may be the best in identifying dementia cases, but under ascertainment is very much a limitation for GP records as well. Furthermore, access to GP records for secondary use is still limited compared to mortality data and hospital admissions.

We have added the following paragraph in our discussion in response to the comment above 'Due to financial incentivisation, there has been an increase in dementia diagnosis in primary care, although concerns of overdiagnosis have been raised and should be monitored carefully. [6] Diagnosis of dementia is usually initiated in a primary care setting and could be considered to be the most complete single source of dementia case ascertainment, but underdiagnosis still exists [16,33,34] for various reasons including reluctance of GPs to diagnose dementia. [34]'

4. Figure 3 is the main crux of the paper. Is there any way it could be drawn so that the size of the areas in the Venn correspond to the number of cases? This would make it easier to interpret visually.

Authors response: Thank you for this suggestion. We have now modified Figure 3 so that the that the size of the areas in the Venn better correspond to the number of cases for each of the data sources.

Reviewer: 2
Dr. Kelvin Jordan, Keele University

Comments to the Author:

Studies in dementia are increasingly using electronic health records, however concerns are raised about whether cases are missed giving the complexity of diagnosis and recording. This study compared primary care, secondary care, mortality and mental health registers on diagnosed cases of dementia. The strengths are the linkages across these datasets including MHDS. The weakness is the low number of practices included.

Authors' response: We thank Dr Jordan for his comment. We refer Dr Jordan to our response above on regarding the small number of GP Practices included. Our main message was to demonstrate (even in this smaller study), the high proportion of missingness which has the potential to impact the reliability of large primary care databases that are increasingly being relied for outcome ascertainment in large studies. Due to the size of the study, we were able to carry out the detailed review of the medical records in GP practice and get first-hand insight into how the coding is handled in GP practices. This would not be possible using primary care databases. We also showed that almost half the dementia cases from GP records were initially missed by the practice and were only ascertained after the confirmation step by the researchers. This is an important finding in terms of missed outcomes of which researchers using such databases should be aware of as this will influence their results. We have tried to make this clearer point in our manuscript to highlight the value of this, regardless of the size of the study.

The text now reads:

'The study in the sub-population of the EPIC-Norfolk cohort with GP records has highlighted several key issues. Even though previous work has shown that practices are confident in identifying dementia cases, [9] the complexity of 'missingness' is well recognised. [32] If we had relied on the initial data extract from practices, we would have missed almost half the dementia cases. These cases were only ascertained via the confirmation step which involved a more detailed interrogation of the medical records by the research team. This may be due to lack of technical capacity or capability and could be a further reason for under-ascertainment. This reveals the potential scale of missed outcomes where known cases are not reflected in extraction of routine records, influencing studies using databases such as CPRD. [33] These findings highlight the importance of extending further training and knowledge of dementia coding in primary care.'

1. The key question it seems to me given dementia is managed in primary care and researchers will generally start with primary care records is how good GP records alone are in identifying dementia. Mortality and hospital records are limited as not all patients will have died and not all patients will attend hospital due to dementia (did you use secondary as well as primary diagnoses?). So the question for the other sources is then really about whether they add patients missed in GP records.

Authors' response: We hope the Venn diagram (Figure 4) helps with this suggestion. There were 10 additional participants with a record of dementia in secondary care records, for whom we were unable to confirm these diagnoses as we could not access their records to check on the detailed information. One additional participant had no indication of dementia or dementia related condition in GP records despite a HES record of dementia. Ideally this would be explored in a larger study, and we have now stated this in the discussion.

'Whether secondary care data add any further to what can be found in GP record could not be explored in our small study. There were 10 additional participants with a record of dementia in cases from secondary care records, although we were unable to confirm these diagnoses as we could not access their records to check on the detailed information. One additional participant had no indication of dementia or dementia related condition in the GP records despite a HES record of dementia. This may be a more widespread issue and should be explored in future work involving a larger and wider range of practices.'

2. Figure 4 seems somewhat reassuring in terms of completeness of GP diagnoses of dementia given the biggest group missed are those who have died who did not still have their records in primary care. This assessment though is weakened by low number of practices taking part so there may be an issue of generalisability, particularly if the best coding practices took part.

Authors' response: We have made it clear that the sub-population was different from the overall cohort and from the general population of older people. None the less, our findings are still widely applicable, and we advise that appropriate consideration to these potential biases.

We include in our Discussion the following statement:

'It is clear from the characteristics and lower dementia incidence that the sub-population was different from the overall cohort with regards to age, education and social class and therefore assuming generalisability to wider populations requires appropriate caveats.'

3. I was slightly confused by P.9, line 10 "Participants with a dementia diagnosis in secondary care, but not indicated in the GP data were further reviewed by researchers in the third and final confirmation phase". What was the evidence they were looking for, if not a Read / SNOMED code, and has this been validated as a means of identifying people with dementia? Was it a free text mention of dementia? This information is important for researchers wanting to use primary care records to know how to obtain all cases of dementia (particularly as free text usually isn't available).

Authors' response: Thank you for highlighting that the protocol was not clear in the original description. This paragraph (which has now been transferred to the Supplementary Information) has now been revised as follows:

'In the second phase, cases identified from GP records were cross-checked with the NHS Digital data, and the level of agreement between the data sources was examined. GP records of individuals with a dementia diagnosis in secondary care but missing from the initial GP data extract were further reviewed by researchers in the third and final confirmation phase of this study. Individual records were reviewed by the researchers for any dementia read or ICD code or mention of dementia as free text. Initial case ascertainment from GPs was conducted between 2016-2017 and NHS Digital data were up to 31st March 2018, with case confirmation in GP records up to 20th December 2018, allowing few months after the follow-up period for the NHS Digital (secondary care) data'

4. It would be useful to know, where there is concordance, what impact does using one source rather than another have on date of diagnosis – which is important for both research using dementia as an exposure or as an outcome. Mortality data obviously would not be used for the former but it is possible dementia was undiagnosed until death

Authors' response: We have now included this information in our manuscript as part of the results:

Where there was concordance in cases for the MHDS and HES data sources (N=448), 71% (319) had a date of diagnosis in the MH dataset before the date of diagnosis in HES, (median (IQR) 0.7 years (0.3, 1.7 years) with 26% (116) with an earlier date in HES and (median (IQR) 0.6 years (0.2, 1.9 years). Only 3% (13) had the same date recorded in both.

In our discussion we have modified the paragraph discussing the MHDS dataset as follows (changes highlighted in blue text):

'We were also able to draw on the relatively new MHDS from NHS Digital, as yet not widely used in research or described in detail. Where there was concordance in cases for the MHDS and HES data sources, we found that for most cases, the date of diagnosis was earlier in MHDS by just a few months. This is as expected, as the MHDS will be from referrals made from GP, where diagnosis of dementia is usually initiated by family or individuals themselves. [6]. All activity relating to patients receiving care for a suspected or diagnosed mental health, learning disability, or neurodevelopmental conditions is within scope of this dataset.'

5. Is concordance of dementia diagnosis across data sources improving over time? Recruitment goes back to 1993 when recording quality was presumably lower.

Authors' response: We did observe a steady increase in recorded incidence of dementia from 1996 (when linkage with health records began) up until 2018 (in contrast to population based studies that report drop in incidence so this is a recording phenomenon as the reviewer notes). This increase was observed in all the data sources and across all age-bands, although most clearly in the over 80s. As others have reported, this is likely to be reflecting an increased awareness and diagnoses of dementia. There was no clear indication of increased diagnoses on implementation of the major dementia policies introduced since 2006 with the increase being more gradual. However, examining effects of policy is not always so straightforward to interpret. Changes in policies take time to implement, and there is almost always delay in uptake. We did not include these findings here as we felt this was beyond the scope of this paper and may address in more detail in future.

6. I understand why the researchers did not explore sub-types of dementia but it is worth discussing a) whether issues affect specific types more, b) potential for people to be recorded with different types over time, or the incorrect type.

Authors' response: Misclassification between dementia subtypes is quite common. We did not analyse types of dementia separately to maximise our outcome by minimising risk of misclassification. Using this approach of a broader is considered to be appropriate as many risk factors are shared, particularly for clinically diagnosed AD and VaD, and mixed which is the most common type in people aged 80 and above. We have now changed the discussion as below to address this point

To maximise our outcome and minimising risk of misclassification, we did not analyse types of dementia separately. Misclassification is common, and the emphasis on clinically diagnosed Alzheimer's disease (AD) in the majority of individuals hides the fact that most people with dementia, namely those aged 80 and above have mixed pathologies in their brains. Those who die without dementia also have increasing expression of AD pathologies [25,26]. Treating dementia as a single entity at a population level is appropriate as many risk factors are shared. Misclassification may have some impact on associations for younger onset dementias, where the level of importance of this will depend on the purpose of the study. Using a broader classification also improves external validity. [27] Under-ascertainment is inevitable and should be considered when making prevalence estimations.

7. I wonder if some codes have been missed eg E00.. Senile/presenile dementia; F116. Lewy body disease; F111. Pick's disease

Authors' response: We thank Dr Jordan for highlighting these codes that have not been included in our list of codes as presented in Supplementary Table 1. We have examined the dataset and found that there are no cases with these codes. We have however, included these additional codes in Supplementary Table 1 and have resubmitted in the revised supplementary information

8. Table 1 – why don't the numbers for men and women with dementia diagnosis within an age band sum to the total for that age band?

Authors' response: We thank the reviewer for highlighting this error. It seems that there was an error for the numbers in the cross-tabulation for the women (this may be due an error in including the filter to include the cohort date up to the end of December 2018). This has now been corrected and the numbers for the age band for men and women separately now add up to the total

9. I don't think figure 2 adds anything and could be deleted.

Authors' response: We would like to retain this Figure as it provides a good visual representation of the findings across ages from mid to later life, a crucial area that is often neglected.

9. I was not sure of the purpose of the analysis behind table 3. Socio-demographic characteristics associated with dementia seems a different question to the underling question here around differences between the sources.

Authors' response: Respectfully we would disagree. There is an abundance of evidence on differential and inequitable access to services and help seeking behaviours that are influenced by far more than health itself. It is therefore important that we look for such influences on the presence or absence in different data sets e.g., the middle classes might differentially access memory clinical psychiatry services early in the dementia process whereas in others dementia might be diagnosed once frail and multimorbid in hospital and close to death. We think Table 3 is important as it is a comparison of the main data sources used in this study. The table shows that for socio-demographic factors associations were similar across the three data sources. This should provide reassurance to researchers who have access to these as their sole data source. We also explain the age differences observed across the three data sources which was as anticipated with younger individuals with memory concerns more likely to be referred to memory clinics and older individuals coming through hospitals.

We explain this in our manuscript in the discussion

'Using simple socio-demographic factors to examine associations with future risk of a definite dementia diagnosis, we demonstrate small differences, across the main two data sources, HES and mortality, with more difference observed in the MHDS. The age differences observed across the three data sources is as anticipated, as younger individuals with memory concerns are more likely to be referred to memory clinics as compared to older frailer individuals who are more likely to be identified through hospitals.'

VERSION 2 – REVIEW

REVIEWER	Charles Marshall Queen Mary University of London, Preventive Neurology Unit
REVIEW RETURNED	29-Apr-2022

GENERAL COMMENTS	All my comments have been addressed well.
---

REVIEWER	Kelvin Jordan Keele University, School of Primary, Community and Social Care
REVIEW RETURNED	10-May-2022

GENERAL COMMENTS	The authors have responded appropriately to my comments
---